# Comparative Experimental Investigation on Optimal Parametric Array Types

**DOI:** 10.3390/s21155085

**Published:** 2021-07-27

**Authors:** Donghwan Jung, Jiyoung Song, Jeasoo Kim, Jaehyuk Lee

**Affiliations:** 1Underwater Vehicle Research Center, Korea Maritime and Ocean University, Busan 49112, Korea; wjdehdghkss@gmail.com; 2Department of Ocean Engineering, Korea Maritime and Ocean University, Busan 49112, Korea; jysong.ualab@gmail.com; 3Vibration & Noise R&D Department, Daewoo Shipbuilding & Marine Engineering Co., Ltd., Geoje-si 53302, Korea; LEEHEL7@dsme.co.kr

**Keywords:** parametric array, dual-frequency parametric array, broadband parametric array

## Abstract

As a sound transmitting device that relies on the nonlinearity of a medium, a parametric array (PA) can generate high-directivity low-frequency signals using a small aperture transducer and high-frequency signals. Despite their relatively low source level, the PA is frequently used to measure the acoustic properties of materials in low-frequency regions owing to their high directivity in confined acoustic water tanks. Therefore, methods for improving the source level of secondary signals are of interest. Currently, there are two driving methods for PA: the dual-frequency PA and the broadband PA with amplitude modulation. In this study, we share the results of an elaborate and comparative experimental investigation of these two driving methods. Comparisons are made and discussed in terms of the intensity of the generated secondary signal and its characteristics in the frequency domain. Based on these factors, we confirmed that the broadband PA was more suitable as the sound source of the low-frequency characteristic measurement system of acoustic materials.

## 1. Introduction

The acoustic characteristics of acoustic materials must be accurately predicted and measured to evaluate the underwater acoustic performance of equipment or objects. These characteristics can vary dramatically depending on temperature and pressure (depth) [1]. Specifically, they are frequency dependent. Therefore, it is necessary to evaluate them over a wide range of frequencies. There are techniques for determining properties, such as the reflection loss and the transmission loss, from the measurements made on panels of the material under test [2,3,4]. Such measurements are generally carried out through experiments in water tanks. However, the measurement of acoustic properties via panel testing in a water tank incurs signal interference problems, such as reflected signals from walls and diffraction signals caused by the finite size of the panel. Such problems are more evident when low-frequency sound sources with a broad beam width are used. To narrow the beam width, the size of the sound source must be increased. However, this approach results in difficulties during operation and physical limitations [5]. Essentially, as the size of the sound source increases, the cost of producing the sound source increases as well, thereby making physical operation difficult.

These limitations and problems can be overcome using a parametric array (PA). The high directivity of a PA allows for the measurement of low-frequency acoustic characteristics and reduces the problems caused by interference signals, such as diffraction signals from the edge of the panel and wall reflection signals [6,7,8]. As a PA generates low frequencies through high-frequency nonlinear interaction, the size of the sound source is small, and the virtual secondary sound source through nonlinear interaction has a narrow beam width, which is similar to that of an end-fire array.

Although a PA can be used to measure the low-frequency acoustic properties of materials in a limited water tank, it has the disadvantage of generating relatively low sound source levels for low-frequency signals, i.e., secondary signals generated using PAs [9]. Therefore, methods for improving the source level of secondary signals are of interest. We tried to improve the source level of the secondary signal by applying an appropriate parametric array type as the system sound source. Since the system to be developed is intended to be used in a water tank, it is necessary to consider issues such as the interaction range of the parametric array and the pseudo sound generated when using the parametric array at a short distance.

In 1975, Merklinger theoretically demonstrated that the efficiency of PA could be improved through suitable periodic shaping of the envelope of the primary wave [10]. In 1979, Moffett confirmed that the measurement value of the secondary wave generated through the broadband PA at a specific point was consistent with the theory [11], and in 1981, theoretically organized the intensity change of the secondary wave according to the interaction range [12]. In addition, various studies have been performed on PA, but these are mostly theoretical studies and experimental studies have been performed on the far field. In this study, two representative types of PAs were investigated, and a comparative experiment was performed to determine the sound source of the low-frequency acoustic property measurement system for use in relatively close range.

The remainder of this paper is organized as follows: The fundamental principles of a PA are introduced in Section 2, including a theoretical model of the PA. This is followed by the description of the comparative experiment in Section 3. The results of the comparative experiment are discussed in Section 4. Finally, the conclusions are presented in Section 5.

## 2. Fundamental Principle of the Parametric Array

In linear acoustics, the change in the volume of the medium depending on the change in pressure is assumed to be linear, and the phase velocity of the sound wave is assumed to be constant. Therefore, when two sound waves with different frequencies propagate in the same medium, they do not interact with each other, and they are considered to be a linear superposition of both waves. However, in nonlinear acoustics, the volume change rate of the medium varies depending on the increase or decrease in the sound pressure, and the phase speed varies for each part of the waveform [13]. In the waveform, the phase speed is high for the part with high sound pressure and is low for the part with low sound pressure. Such changes in phase velocity appear to increase as sound pressure increases. Additionally, as the sound wave of high sound pressure progresses, the waveform is gradually distorted to generate additional frequency components. In the case where a sound wave with high sound pressure is a sine wave, its waveform is gradually distorted and transformed into a shock wave. Therefore, in nonlinear acoustics, when two waves move in the same direction, each wave propagates in a medium along with the disturbance of the other wave, and scattering through sound occurs because of the non-uniformity of the medium [14]. Thus, two sound waves with different frequencies interact and distort each other owing to the nonlinearity of the medium, thereby generating a difference frequency and a harmonic component.

However, in general, the higher the high frequency, the greater is the attenuation in the medium. Thus, high-frequency components are dissipated during propagation and only the difference frequency component remains. As such, generating a low-frequency signal using the nonlinear interaction of high-frequency signals is called a PA, as proposed by Westervelt [15].

### 2.1. Dual-Frequency Parametric Array

Two types of PA exist. The first one involves a dual-frequency PA in which a difference frequency signal with a single frequency is generated using two single-frequency signals as the primary wave. Westervelt [15] proposed the case of an absorption-limited array in the near-field, which assumed that the primary wave is a plane wave in which two high frequencies with a narrow beam width are highly collimated. Since then, many studies have been conducted, such as the beam pattern of the PA using the two signals, the calculation of the intensity of the generated low-frequency signal, and the characteristics at a short distance [16,17,18,19,20,21,22,23]. In particular, Berktay and Leahy [24] proposed a model that explains the three-dimensional characteristics of the interaction domain, considering the spreading loss and directivity of both primary waves. This model presents the following results assuming that the interaction occurs at the far-field of the sound source and the sound source is placed in the y–z plane, where the sound axis coincides with the *x*-axis [5] at the coordinates shown in Figure 1.
(1)$$P_{d}\left(R,\theta ,\eta \right)=\frac{\omega_{d}^{2}p_{1}p_{2}\beta }{4\pi \rho c^{4}R}e^{-\left(\alpha_{d}+ik_{d}\right)R}\int_{-\frac{\pi }{2}}^{\frac{\pi }{2}}\int_{-\frac{\pi }{2}}^{\frac{\pi }{2}}\frac{D_{1}\left(\gamma ,\phi \right)D_{2}\left(\gamma ,\phi \right)\cos\gamma }{\alpha_{T}+ik_{d}\left(1-u\right)}d\gamma d\phi \;,$$
where $P_{d}$ represents the secondary wave pressure and $p_{1}$ and $p_{2}$ represent the primary wave pressures. Additionally, $\alpha_{T}=\alpha_{1}+\alpha_{2}-\alpha_{d}$, where $\alpha_{1}$ and $\alpha_{2}$ represent the attenuation coefficients of the primary wave, and $\alpha_{d}$ represents the attenuation coefficient of the secondary wave. ρ represents the density of the propagation medium, β represents the nonlinear coefficient, $\omega_{d}$ represents the angular frequency of the second wave, c represents the speed of sound, $k_{d}$ represents the wave number of the secondary wave, and  u=cosγcosθcos(ϕ−η)+sinγsinθ; $D_{1}$ and $D_{2}$ represent the directional functions of the primary wave.

### 2.2. Broadband Parametric Array

If the primary wave is a broadband signal, an infinite number of frequency components interact to generate a broadband secondary wave, which is called a broadband PA. In 1965, Berktay [25] extended the PA theory to a broadband signal by proposing a far-field solution based on a quasi-linear approximation of the Westervelt equation. Berktay’s far-field solution assumes that the primary wave is a signal in which a single frequency carrier is modulated using the envelope function, which is a plane wave, and moves along the acoustic axis. Berktay’s approximate solution in the far field is expressed as follows [26]:(2)$$p_{2}\left(R,t\right)=\frac{\beta p_{0}^{2}S}{16\pi \rho c^{4}R\alpha_{1}}\frac{\partial^{2}}{\partial \tau^{2}}\;E^{2}\left(\tau \right),$$
where $p_{0}$ represents the pressure of the primary wave, E(τ) represents the envelope function, S represents the cross-sectional area of the sound source, and τ=t−R/c represents the retarded time. For beta in Equations (1) and (2), 3.5 ± 0.1 is generally used. This is determined for water at room temperature, which is in good agreement with the isentropic equation of the state and the results of the finite amplitude method [27]. Equation (2) shows that the demodulated secondary wave has a waveform of the second derivative of the square of the envelope, and the amplitude is proportional to the square of the initial pressure $p_{0}$. After Berktay’s proposal, interest in broadband PAs grew and a number of related studies were conducted [10,11,28]. From the results of these studies, broadband PAs are known to improve the efficiency of PAs by 2.1–6.0 dB compared to dual-frequency PAs [10]. Furthermore, the dual frequency PA needs to be produced as a special sound source to ensure that two high-frequency signals of strong output are collimated, while the broadband PA has the advantage of being a high-frequency sound source that can output a strong signal. Recently, studies have been conducted on a modulation technique of primary signal of broadband PA [29,30,31].

## 3. Experimental Setup

Experimental measurements were carried out in a 2.5 m long × 1.2 m wide × 1.2 m deep acrylic tank, which was filled with fresh water to a depth of 1.1 m. The water temperature was monitored and controlled to remain constant. An acoustic source transducer was placed at one end of the tank at a depth of approximately 0.5 m, and a hydrophone was mounted at the same depth. A truncator (acoustic filter) was also mounted in the tank for some experiments.

The acoustic source was a BII-7544 (Benthowave Instrument Inc., Collingwood, ON, L9Y 0B4, Canada) made for the dual-frequency PA, which can output two primary waves. This had an output transmitting voltage response (TVR) of 173 and 170 dB re 1 uPa/V at 1 m at 200 and 195 kHz respectively, and a −3 dB beam width of 6° at 200 kHz and 195 kHz.

Pulse generation and data acquisition were performed using the PXI system (NI), and the generated pulse waveform was amplified through an E&I 1004 power amplifier, which was used to drive the transducer. A Bruel and Kjaer 8103 omnidirectional hydrophone was used as a receiver. The signals received by the hydrophone were amplified using a Bruel and Kjaer Nexus charge amplifier, which had a 10–22.4 kHz band-pass filter and a 22 dB gain on its output (Figure 2). In order to enhance the signal-to-noise ratio, the coherent averaging over 50 independently measured time series was used. Then, the Fast Fourier transform (FFT) analysis was carried out using MATLAB to obtain the spectral information.

For the comparison experiments, the pulsed waveform *w*(*t*) used in the broadband frequency PA is a high-frequency carrier wave. That was amplitude modulated using a low-frequency raised cosine bell envelope *E*(*t*), as follows:(3)$$w\left(t\right)=E\left(t\right)\sin\left(2\pi f_{p}t\right)\;\;,\;\;\;\;0<t<1/f_{e}$$
(4)$$E\left(t\right)=\left(1-\cos\left(2\pi f_{e}t\right)\right)/2\;,\;\;\;\;0<t<1/f_{e},\;$$
where $f_{p}$ represents the primary wave frequency of the transducer and $f_{e}$ represents the envelope frequency. The reference for the low frequency component for comparison in the experiment is 5 kHz, and for this purpose the values used for this pulse waveform were $f_{p}$ = 200 kHz and $f_{e}$ = 5 kHz. To generate the same low frequency component of 5 kHz, the pulses used for the dual frequency PA were sine waves of 195 kHz and 200 kHz, with a length of 0.2 ms to equal the broadband PA.

The truncator used in this work is made from a 6 mm thick stainless steel panel. The thickness of the truncator is designed to strongly attenuate the 200 kHz primary waves and to transmit the secondary waves with little attenuation. In a small size water tank, where the distances are short and there is limited absorption of the propagating waves, a truncator is useful for terminating the region of non-linear interaction in a PA by physically filtering out the high-frequency primary component and allowing the low-frequency secondary waves through. This is necessary to avoid the nonlinear generation of secondary signals in the region of the test panel, where the measurement is performed [32]. Truncation of the PA is also advisable to avoid pseudo sound caused by the nonlinearity in the receiving hydrophone or electronics [7,32]. Even with truncation, the narrow beam characteristics at low frequencies make a PA a useful source in confined laboratory conditions [33]. The truncator was approximately 0.6 m × 0.6 m × 0.006 m in size. The size of the truncator in these experiments was selected by considering several factors. One factor was the requirement to avoid diffractions from interfering with the direct signals in the experiment. As the truncator’s beam width was 6° at 200 kHz, the 0.6 m × 0.6 m panel avoided any diffraction that could have occurred at the edge of any location in the tank. Theoretically, the truncator resulted in a greater than 24 dB reduction in the signal at the primary frequency (200 kHz) versus a –3 dB reduction at the secondary frequency (5 kHz).

The comparative experiment involved two measurement experiments named Ex(1) and Ex(2) depending on the presence or absence of a truncator. To observe the change in the secondary wave according to the interaction range, in Ex(1), as shown in Figure 3, the distance between the receiver and the sound source was adjusted from 0.4 m to 1.8 m at 0.2 m intervals, after which the primary and secondary waves were measured. In Ex(2), a truncator that attenuates the primary wave was fixed at 0.2 m from the receiver to reduce the effect of the pseudo sound caused by the strong primary wave. As shown in Figure 4, with the truncator fixed at a specific distance from the receiver, the distance between the sound source and the receiver was changed from 0.4 m to 1.8 m at 0.2 m intervals as in Ex(1), after which the primary and secondary waves were measured. The depth and relative distances of the transmitter and receiver were established to avoid unwanted sound reflections by ensuring that the direct and reflected waves do not overlap.

## 4. Results and Discussion

Measurement Results of Comparative Experiment

The time-series data of the measurement results obtained based on the distance between the sound source and the receiver in Ex(1) are as follows: Figure 5a shows the results of the broadband PA; Figure 5b shows the results of the dual-frequency PA. In the data acquired at each receiver position, the dotted line portion with a length of 0.32 ms represents the direct wave portion.

The time-series data of the measurement results obtained based on the distance between the sound source and the receiver in Ex(2), equipped with a truncator in front of the receiver, are as follows: Figure 6a shows the results of the broadband PA, and Figure 6b shows the results of the dual-frequency PA. Similar to the measurement results of Ex(1), the dotted line portion of 0.32 ms in the data collected at each receiver position represents the direct wave portion.

Figure 7a,b show the corresponding power spectra, which were calculated from the direct wave signals among the time-series data of Ex(1), displayed in the plots for Figure 4b and Figure 5a. Figure 7a,b show the corresponding power spectra calculated from the direct wave signal, which is the dotted line part of the time-series data of Ex(2) shown in the plots for Figure 5a,b. By comparing Figure 7 and Figure 8, it can be confirmed that the intensity of the primary wave is attenuated by approximately 24 dB in Ex(2) compared to Ex(1). This is the same value as the theoretical attenuation of the truncator, which means that the primary wave is attenuated by the truncator. From the results of the measured data in the frequency domain, it can be observed that the intensity of the primary wave, 200 kHz, is similar in both the broadband and dual-frequency PAs. However, there is a difference in the intensity of the secondary wave, which has a low frequency. In addition, in the case of the broadband PA, it can be observed that the frequency bands of the primary and secondary waves are relatively wide. Figure 9 shows a plot of the intensity of the first and second waves calculated for the direct wave in each comparative experiment depending on the distance between the transmitter and the receiver. The plotted results are corrected for 22 dB amplification of the receiving amplifier and 38 dB attenuation of the 200 kHz component owing to the band-pass filter. Figure 10 shows the comparison of Ex(1) and Ex(2) for easy confirmation.

## 5. Discussion

Our goal is to find a suitable PA type for using the PA as the sound source for the measurement system in a tank that is relatively close distance. To this end, we conducted an experiment and compared the two types of PA, where we studied the effect of the interaction range on the intensity of the secondary wave, the ability of discriminate of secondary waves and the influence of pseudo sound. In Ex(1), without a truncator, a strong first-order wave signal was received, thereby generating a pseudo sound. From the results of Ex(1), the power of the primary wave of both PAs was similar at all receiver locations. However, for the secondary waves, the amount of attenuation in the broadband PA differed from that of the dual-frequency PA as the receiver moved away from the sound source. In the case of the broadband PA, the attenuation of the secondary waves’ intensity decreased compared to that of the dual-frequency PA. Considering that the amplitude of the pseudo sound is proportional to the square of the amplitude of the primary wave, it can be inferred that the effect of the pseudo sound is relatively small in the broadband PA compared to the dual-frequency PA.

In the results of Ex(2) in which the effect of the pseudo sound was removed by attenuating the intensity of the primary wave through a truncator, it can be confirmed that the intensity of the secondary wave generated by the broadband PA was stronger than that of the dual-frequency PA. In addition, the frequency domain results according to both PAs showed distinct differences. It can be observed that the spectrum results of the broadband PA in the 20 kHz–180 kHz band show a significantly lower intensity compared to that of the secondary wave band or 200 kHz, which is the center frequency of the primary wave. Conversely, in the case of the dual-frequency PA, it can be observed that the intensity of the 20 kHz–180 kHz band is stronger than that of the broadband PA because of the side lobe. This means that broadband PAs not only generate stronger secondary waves than dual-frequency PAs but also have better discrimination for secondary waves.

## 6. Conclusions

Comparative experiments on two PAs were performed in a small tank. The experiments focused on whether the pseudo sound was influenced by the presence or absence of a truncator; the direct wave was separated from the received data and analyzed. In all the experiments, the power of the primary wave of both PAs was similar at all receiver positions.

The comparative experimental results showed that the broadband PA had a lower influence on the pseudo-sound than the dual-frequency PA, and that the broadband PA produced a 3–4 dB stronger secondary wave compared to that of the dual-frequency PA. In addition, compared to the dual-frequency PA, the broadband PA had a wider band of secondary waves and a lower spectrum value between the primary and secondary frequencies, and thus, the secondary waves were easily identified. In dual-frequency PA, a study was conducted to develop a lower frequency broadband collimated and steerable acoustic beam source [34]. However, in general, dual-frequency PA generates a secondary wave of a narrow band and special manufacture is required for the secondary generation of a wide band, whereas broadband PA is possible through modulation of the primary wave. Based on the experiments’ results, it was confirmed that the broadband PA, which has stronger secondary wave intensity and improved discrimination, is suitable as a sound source of a system for measuring the low-frequency characteristics of acoustic materials. In this study, the dual frequency PA and the broadband PA were driven using the same sound source, and the stronger secondary wave intensity by the broadband PA observed corroborates with the results of Merklinger’s theoretical study.

This study started with constructing a system to measure the low-frequency acoustic properties of sound source materials in a limited-sized water tank. A high directional sound source is required to solve various interference signals in a tank of limited size, and a comparative experiment was performed to use PA that satisfies this as a sound source. For the numerical simulation of the sound field of a parametric array, it is necessary to calculate the volume in which the interaction takes place, but it is relatively difficult to calculate the volume due to the influence of the near field in a limited-sized tank environment. Although various studies are being conducted, such as research on the development of low-frequency broadband collimated sound sources, in this study, when using general broadband PA and dual-frequency PA, experiments were performed and showed results as the first step in using PA to measure acoustic characteristics at limited distances. In our future studies, we shall focus on the construction of a system for measuring the low-frequency acoustic properties of acoustic materials through experiments on the envelope function and the distance between the acoustic material and the sound source.

## Figures and Tables

**Figure 1 sensors-21-05085-f001:**
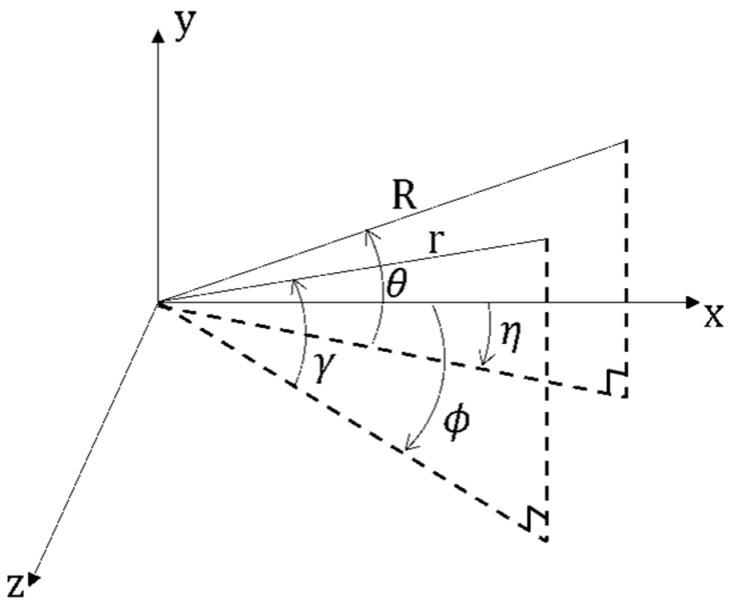
Geometry for the analysis of the Berktay and Leahy model (field point (*R*, *θ*, *η*), source point (*r*, *γ*, *ϕ*)).

**Figure 2 sensors-21-05085-f002:**
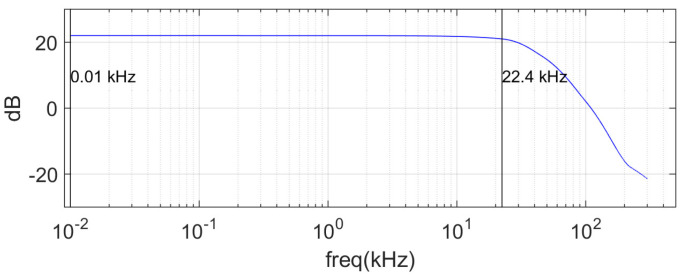
Bandpass filter and gain set in amplifier.

**Figure 3 sensors-21-05085-f003:**
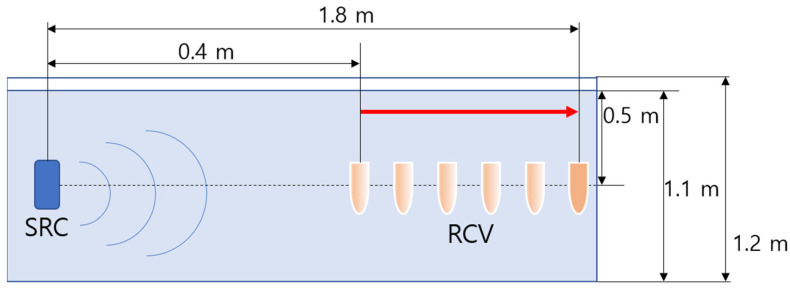
Schematic diagram for configurations associated with Ex(1).

**Figure 4 sensors-21-05085-f004:**
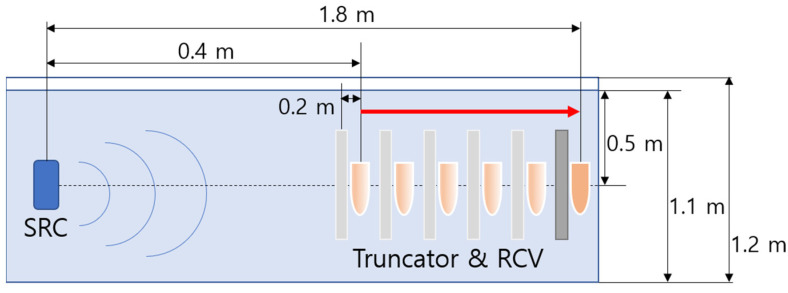
Schematic diagram for configurations associated with Ex(2).

**Figure 5 sensors-21-05085-f005:**
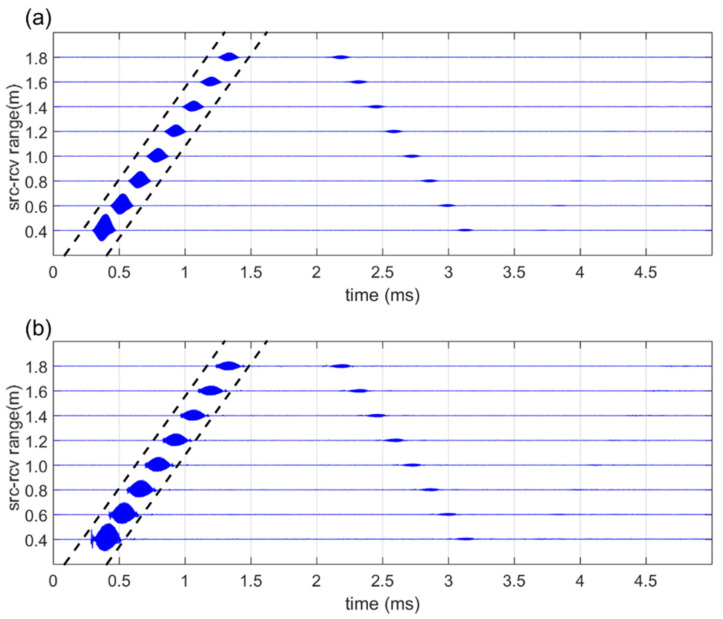
Received signal for Ex(1): (**a**) Broadband PA; (**b**) Dual-frequency PA.

**Figure 6 sensors-21-05085-f006:**
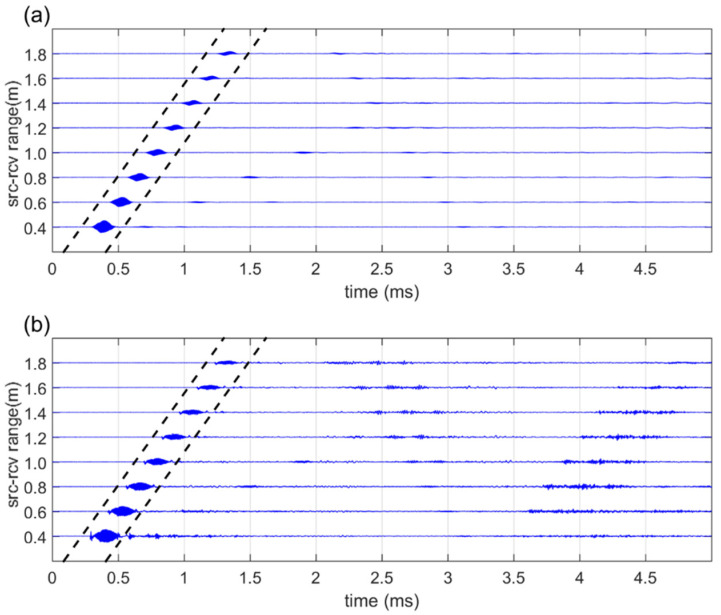
Received signal for Ex(2): (**a**) Broadband PA; (**b**) Dual-frequency PA.

**Figure 7 sensors-21-05085-f007:**
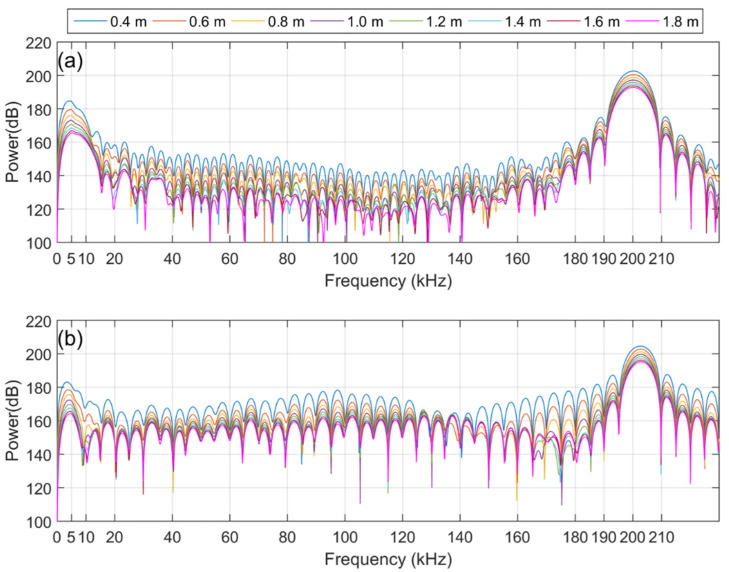
Received signal for Ex(1): (**a**) Broadband PA; (**b**) Dual-frequency PA.

**Figure 8 sensors-21-05085-f008:**
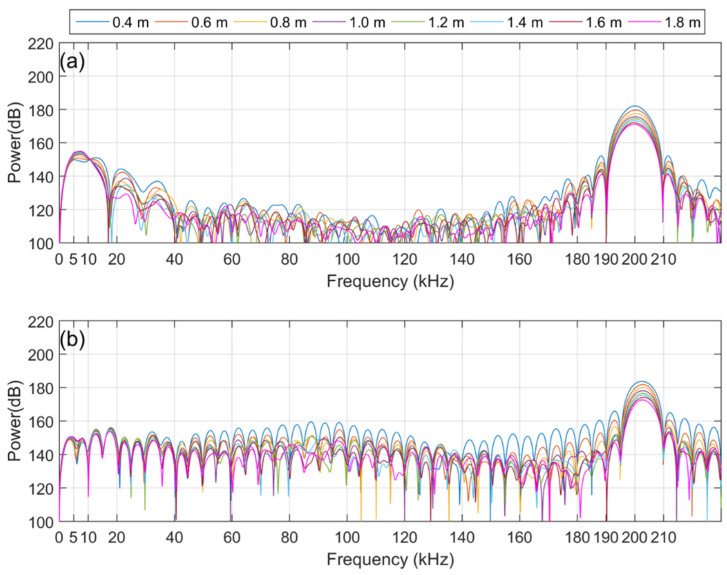
Received signal for Ex(2): (**a**) Broadband PA; (**b**) Dual-frequency PA.

**Figure 9 sensors-21-05085-f009:**
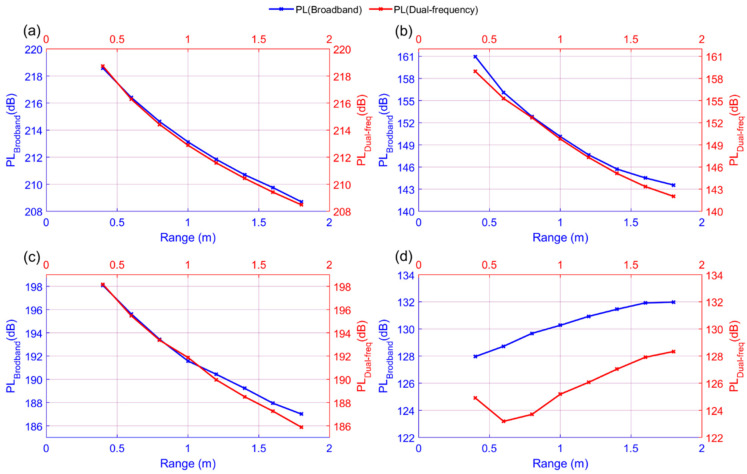
Sound pressure level of the primary and secondary waves depending on the distance between the source and the receiver for broadband and dual-frequency PAs: (**a**) Primary frequency in Ex(1); (**b**) Secondary frequency in Ex(1); (**c**) Primary frequency in Ex(2); (**d**) Secondary frequency in Ex(2).

**Figure 10 sensors-21-05085-f010:**
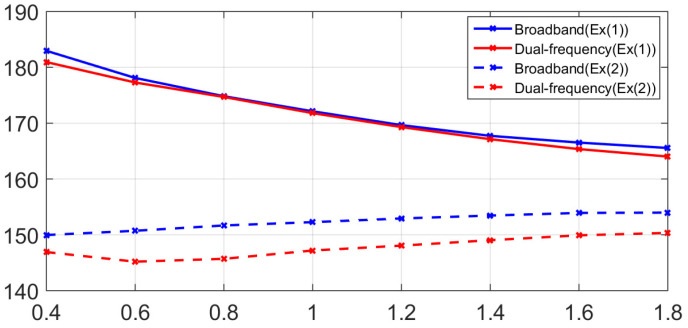
The intensity of the secondary wave according to the distance between Ex(1) and Ex(2).

## Data Availability

Not applicable.

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
