# Peer review of "Comparative Experimental Investigation on Optimal Parametric Array Types"

_sensors, 2021, doi:10.3390/s21155085_

Round 1
Reviewer 1 Report
This article (communication) has investigated the effectiveness of each of the dual-frequency and broadband PAs through a comparative experimental study. The contents clearly show the characteristic difference of the acoustical performance from both frequency- and time-domain standpoints.
>>Title
The title is slightly unclear. Please provide a more simple and intrinsic title.
>>English quality
The English may be grammatically and editorially proofread by professional service.
>>
The caption of figure 6 may be modified as "Received signal for Ex(1). (a)...".
Author Response
Thank you for the comment and suggestion of this manuscript and I have revised the content and answered the question.
The reviewed suggestions make me write a more complete paper and thank you very much.
Please see the attachment.

Reviewer 2 Report
The authors present an experimental comparison study of parametric low frequency sound generation in a confined environment. The major concerns with the manuscript are: 1) the novelty of the work must be made clear given the large amount of parametric array literature that already exists, 2) the experiments cover a very specific configuration and a limited parameter range, so they must be supported with predictions in order to help place the experiments in context, and 3) the results and discussion doe not explore the reasons for differences, nor is it explained whether the limited experiments shown here are broadly applicable to a variety of instrumentation and applications that other users may employ. Specific comments and question follow.
L 57. Have direct comparisons of the dual frequency and broadband approaches ever been done? If so, cite them and make clear how the new work is different. If not, clearly state this, as it would demonstrate the novelty of the work.
L130. Before describing the experiments, it would help to discuss some numerical examples for each PA generation technique and use them to describe the design and implementation tradeoffs with each.
L133. Was the water degassed or otherwise treated to control nonlinearities due to gas content? Was water temperature monitored or controlled?
L139. '... 173 and 170 dB ', was stated for TVR, but only one frequency listed. Please clarify. More generally, please clarify what the drive frequencies were and there fore the intended difference frequency.
L156. It appears that the difference frequency for the broadband technique was not the same as for the dual frequency. Please clarify, and if they were different, explain why this was done and how it might impact data analysis.
L184. More detail is needed on the signal processing and specifically what measures will be used to compare PA methods.
Fig 6. The caption should be for Ex(1), not Ex(2), if it follows the text.
Figures 6 and 7 are difficult to compare in the present form, especially when they land on different pages. It is recommended that representative spectra for a single range (e.g. 1m) be extracted and overlaid for Ex1 and Ex2.
L256. For this paper to be impactful to the research community, the authors should explain and discuss why the two PA methods performed differently and whether the differences are significant for the intended work.
Author Response
Thank you for all the comments and suggestions of this manuscript and I have revised the content and answered the question item-by-item (the black font corresponding to the Questions and the red font corresponding to the Answers).
Please see the attachment

Reviewer 3 Report
Specify because the measurements were carried out in water tanks.
How was the measuring system calibrated?
Was the water tank used by other authors in other researches?
You need to insert references.
In the water tank, we can have unwanted sound reflections how are they eliminated?
The measurement method should be detailed in more detail.
Further references in the measurement system should be added.
Only two graphs to explain the results are too few for a scientific paper.
A theoretical comparison with the measurement method should be made.
The discussion is too short and should be expanded in more detail. For example, the aims of the research and possible practical applications should be better explained.
Explain the actual contribution of the authors to the research and what are the possible developments.
Author Response

(The authors gave the same response as above.)

Round 2
Reviewer 2 Report
The authors have made some useful improvements to their manuscript, but some questions in the original review were not addressed and major concerns still remain. Chief among these is that the experiment results were presented for just one embodiment of the concept, and without any theoretical or numerical support, there is no clear way to generalise the results to other configurations. In other words, these results may only be achieved with the specific transducers and tank size employed, and the value of the results to the readers is therefore extremely limited. The authors say that the numerical evaluation of the concept in small tanks is difficult, but this is not a justification for a limited paper. Either free field theoretical results should be presented as a starting point, or additional small tank configurations should be experimentally evaluated so that the critical aspects of the problem can be explained to the reader.
Author Response
Thank you for your thorough review and insightful comments. You suggestions and comments have helped us improve our manuscript. Our detailed responses to reviewer’s comments are provided below in blue.
Please see the attachment.

Reviewer 3 Report
no
Author Response
we would like to thank you for your valuable time.